# Biallelic Loss of 7q34 (*TRB*) and 9p21.3 (*CDKN2A*/*2B*) in Adult Ph-Negative Acute T-Lymphoblastic Leukemia

**DOI:** 10.3390/ijms251910482

**Published:** 2024-09-29

**Authors:** Natalya Risinskaya, Abdulpatakh Abdulpatakhov, Yulia Chabaeva, Olga Aleshina, Maria Gladysheva, Elena Nikulina, Ivan Bolshakov, Anna Yushkova, Olga Dubova, Anastasia Vasileva, Tatiana Obukhova, Hunan Julhakyan, Nikolay Kapranov, Irina Galtseva, Sergey Kulikov, Andrey Sudarikov, Elena Parovichnikova

**Affiliations:** 1National Medical Research Center for Hematology, 125167 Moscow, Russia; patakh1997@mail.ru (A.A.); uchabaeva@gmail.com (Y.C.); dr.gavrilina@mail.ru (O.A.); makislitsyna@gmail.com (M.G.); lenysh2007@rambler.ru (E.N.); julymbo04@gmail.com (I.B.); ann.unikova@bk.ru (A.Y.); doe30102001@gmail.com (O.D.); vasilnastia@yandex.ru (A.V.); obukhova_t@mail.ru (T.O.); oncohematologist@mail.ru (H.J.); immunophenotyping.lab@gmail.com (N.K.); irinagaltseva@gmail.com (I.G.); smkulikov@mail.ru (S.K.); dusha@blood.ru (A.S.);; 2Institute of Biodesign and Modeling of Complex Systems, I.M. Sechenov First Moscow State Medical University, 119991 Moscow, Russia

**Keywords:** biallelic loss, T-ALL, T-cell receptor beta (*TRB*), *CDKN2A*, chromosomal microarray (CMA), *TRB* clonality

## Abstract

Tumor cells of acute lymphoblastic leukemia (ALL) may have various genetic abnormalities. Some of them lead to a complete loss of certain genes. Our aim was to reveal biallelic deletions of genes in Ph–negative T-ALL. Chromosomal microarray analysis (CMA) was performed for 47 patients with de novo Ph–negative T-ALL, who received treatment according to RALL-2016m clinical protocol at the National Medical Research Center for Hematology (Moscow, Russia) from 2017 to 2023. Out of forty-seven patients, only three had normal molecular karyotype. The other 44 patients had multiple gains, losses, and copy neutral losses of heterozygosity. Biallelic losses were found in 14 patients (30%). In ten patients (21%), a biallelic deletion of 9p21.3 involved a different number of genes, however *CDKN2A* gene loss was noted in all ten cases. For seven patients (15%), a biallelic deletion of 7q34 was found, including two genes—*PRSS1*, *PRSS2* located within the T-cell receptor beta (*TRB*) locus. A clonal rearrangement of the *TRB* gene was revealed in 6 out of 7 cases with 7q34 biallelic loss. Both biallelic deletions can be considered favorable prognostic factors, with an association with 9p21 being statistically significant (*p* = 0.01) and a trend for 7q34 (*p* = 0.12) being observed.

## 1. Introduction

Acute T-cell lymphoblastic leukemia (T-ALL) is a type of aggressive hematological cancer that originates from early T-cell progenitor cells. In T-ALL, malignant hematopoietic cells with immature T-cell characteristics infiltrate the bone marrow. This type of leukemia accounts for approximately 20–25% of cases in adults [1,2,3,4]. The study of cytogenetic abnormalities in T-ALL cells is essential for classifying patients into specific risk groups. This information can help determine the overall treatment approach at the beginning of therapy. However, according to Parovichnikova et al. [5], karyotype abnormalities identified at the onset of T-ALL were not associated with improved or worse 5-year survival.

Over the past decade, researchers have conducted systematic screening of T-acute lymphoblastic leukemia (T-ALL) genomes using high-resolution microarray analysis and next-generation sequencing technologies. These studies have revealed that T-ALL is associated with the accumulation of genetic abnormalities that affect T-cell development. For example, deregulation of transcription factors and disruption of the CDKN2A/CDKN2B cell cycle regulators are found to play a significant role in the pathogenesis of T-ALL [2,4]. It has been found that the absence of biallelic deletion of *TCRG* (7p14), the surface expression of CD13, and heterozygous deletions of the short arm of chromosome 17, as well as mutations in the *IDH1*/*IDH2* and *DNMT3A* genes, are associated with a poor prognosis. In contrast, the expression of CD8 or CD62L, homozygous deletions of *CDKN2A*/*CDKN2B*, mutations of *NOTCH1* and/or *FBXW7*, and mutations or deletions of the tumor suppressor gene *BCL11B* have been associated with improved overall survival. It is important to note that the prognostic significance of homozygous *CDKN2A*/*CDKN2B* deletions is limited to cortical and mature T-cell acute lymphoblastic leukemia (T-ALL). Conversely, mutations in *ID1*/*ID2* and *DNMT3A* are specifically associated with poor outcomes in early, immature T-cell ALL in adults [4].

In this study, we focused on biallelic deletions detected by CMA and their occurrence in the population of T-ALL patients. We also analyzed the localization and association of these deletions with response to therapy. Earlier, the analysis of the 3-year relapse free survival (RFS) revealed a significant association with the absence of minimal residual disease (MRD) on day 70. These data allow us to consider day +70 as the most informative time point; the detection of MRD in the bone marrow at this time discriminates patients into a group with an unfavorable prognosis who might benefit from allogeneic HSCT, as the probability of RFS for these patients at this point is significantly lower (47% for B-ALL, 35% for T-ALL) compared to patients with negative MRD values (78% for B-ALL and 79% for T-ALL) (*p* < 0.001) [5]. Therefore, our goal was to identify biallelic deletions and assess the association between the most common ones and MRD status on day 70 after the start of treatment, as well as overall and event-free survival.

## 2. Results

The main characteristics of 47 patients included in the study are presented in Table 1.

In the first stage, we compared the results obtained by the CMA with the results of CCA and FISH performed for patients on RALL-2016 treatment protocol (Appendix A) at the onset of the disease. Usually, when analyzing the tumor genome by CMA, aberrations larger than 5 MB and additional microdeletions involving the *CDKN2A*/*B*, *BTG1*, *EBF1*, *ETV6*, *ERG*, *IKZF1*, *PAX5*, and *RB1* genes are taken into account. Aberrations smaller than 5 MB in size should be considered only if they are associated with known leukemia-associated genes or other neoplasia-associated genes, including cell cycle regulators, oncogenes and tumor suppressor genes [6]. However, since our focus was on biallelic deletions, a combination of two events that could be germinal in origin, we used thresholds for detecting deletions and duplications greater than 50,000 base pairs (bp) and greater than 3 million bp for cnLOH, as proposed for the study of hereditary aberrations [7]. A complete data on the dimensions and localization of the detected aberrations is presented in Appendix A. Of the forty-seven patients, only three had a normal molecular karyotype. The other 44 patients had multiple deletions, duplications, and copy neutral loss of heterozygosity. Biallelic deletions were found in 14 patients (30%) (Table 2). Data on the T-ALL immunophenotype and MRD status on day 70 are also provided in Appendix A.

### 2.1. Biallelic Deletions in the Study Cohort

Ten patients (21%) had a biallelic deletion of 9p21.3 involving a different number of genes, but in all cases it included the loss of the *CDKN2A* gene. The combination of events leading to the complete loss of fragment 9p21.3 is shown in Table 2. In six cases, overlapping deletions led to complete allele loss, in one case, duplication of a deletion was due to copy neutral LOH, and in three cases, biallelic deletion was the result of two aberrations whose boundaries coincided. We also found monoallelic deletions of 9p21.3 in patients 7, 11, 19, 21, 39, and 40 (Appendix A). In patients #7 and #21, the deletions occurred within the cnLOH region. It is possible that the two aberrations occurred in reverse order, resulting in a monoallelic deletion rather than a biallelic variant.

Seven patients (15%) had a biallelic 7q34 loss (Table 2). Only in one case this was a result from overlapping deletions of different lengths (patient #14). In other cases, the boundaries of deletions on homologous chromosomes matched. Additionally, we found a monoallelic deletion in seven other patients (patients 1, 5, 21, 24, 28, 31, and 32). The location and size of this deletion was approximately 126,000 base pairs, similar to the biallelic deletions in other patients. This deletion includes only two genes—*PRSS1*, *PRSS2*, a part of the trypsinogen cluster located at the T-cell receptor beta (*TRB*) locus. In one patient (#27), the biallelic loss identified by CMA was combined with the t(6;7)(q23;q34) translocation, revealed by CCA. In the remaining patients with bi- and monoallelic deletions, no additional 7q34 aberrations were found (See Supplementary Material Appendix A).

Four patients had both biallelic deletions of 7q34 and 9p21.3 (Figure 1).

Patient No. 26 had three biallelic deletions: additionally, deletion of 13q14.2, which included the *RB1*, *LPAR6*, and *RCBTB2* genes, was revealed. Patient # 30 had a deletion of 12p13.2 that resulted in the complete loss of the *ETV6* gene. No other biallelic deletions were found in any of the patients.

### 2.2. Association between 7q34 Loss and Clonal TRB Rearrangement

T-ALL is characterized by T-cell clonality associated with TRG gene (7p14) rearrangement, as well as TRB gene (7q34) rearrangement in some cases. It is likely that the clonal rearrangement appears as a deletion on CMA. Therefore, we tested TRB clonality for 47 patients using the same DNA samples taken on CMA. Examples of clonal variants associated with biallelic 7q34 loss are shown in Figure 2.

In the group of patients with 7q34 biallelic loss, the majority of patients had TRB clonality, except for one patient. The same result was obtained in the group with monoallelic deletion. Groups without 7q34 genetic abnormalities (33 patients) include patients with polyclonal result (23 patients) and TRB-clonality (10 patients) (69.7% vs. 30.3%). The data are presented as a histogram in Figure 3a.

We hypothesized that patients with a normal 7q34 locus and *TRB* clonality might have smaller deletions than the threshold of 50 kb we chose for deletions. We analyzed CMA data for all samples in our cohort, with a minimum threshold of 1 kb or less for deletions. We found additional seven biallelic and four monoallelic deletions of 7q34 of 5 kb or more in eleven patients. However, these findings did not confirm a direct association between *TRB* clonality and deletions of 7q34. As a result, 10 of the 25 patients (40%) with confirmed biallelic or monoallelic deletion of 7q34 had no *TRB* clonality, while 6 of the 22 patients (27.3%) with a normal 7q34 locus did have this clonality. The data are presented in Figure 3b and Appendix A.

### 2.3. Association between 7q34 and 9p21.3 Losses and MRD Status at Day 70

We used MRD status on day 70 as a surrogate endpoint. This choice was based on the fact that some patients received treatment according to the RALL-2016 protocol, which does not include additional targeted therapy, and some patients were treated under a modified version of the RALL-2016 protocol that includes targeted therapy for patients with MRD. Up to day 70, all patients were treated according to a unified protocol, there were no cases of early mortality before day 70. We analyzed the association between biallelic and monoallelic loss and MRD status on day 70. One patient was transplanted before the 70th day and her MRD status was not determined on the 70th day, so she was excluded from the analysis. The frequency analysis of the association of MRD status on day 70 and the presence of deletion 7q34 (*p* = 0.1262) and 9p21.3 (*p* = 0.0119) is illustrated by Figure 4.

Following MRD detection on the 70th day, patients 3, 5, 6, 16, 20, 21, and 29 were transferred to other treatment protocols due to complications and for medical reasons according to the decision of the expert commission. Therefore, it is difficult to estimate the overall survival rate (Figure 5) accurately. We did not find any significant differences in patients’ outcomes depending on the presence of deletions. This may be due to the fact that a number of patients received targeted therapy or allogeneic HSCT in the case of MRD positivity.

## 3. Discussion

Cytoband 9p21.3 is expected to play a significant role in the formation of biallelic deletions. Cyclin-dependent kinase inhibitor 2A/B (*CDKN2A*/*B*) genes are frequently altered in acute lymphoblastic leukemia (ALL) patients. Data on the prognostic value of the *CDKN2A*/*B* deletion are contradictory. Zhang et al. showed by meta-analysis that *CDKN2A*/*B* deletions were independent poor prognostic markers for both adult and pediatric ALL patients [8]. According to Wang et al. the most common type of *CDKN2A* deletion was homozygous loss and adult T-ALL patients with *CDKN2A* loss had a poor prognosis. These patients might benefit from intensive chemotherapy or allogeneic hematopoietic stem-cell transplantation [9]. Remke et al. noted when compared within child patients with 9q21.3 balanced leukemias, neither heterozygous nor homozygous deletions of the *CDKN2A* gene locus were associated with a differential treatment response [10]. According to VanVlierberghe homozygous loss of *CDKN2A*/*CDKN2B* is associated with a favorable outcome [4]. We have also shown that biallelic 9p21.3 loss can be considered as a statistically significant (*p* = 0.0119) favorable prognostic factor. 

We have noted a surprisingly high incidence of 7q34 biallelic deletions and their repeated localization in our cohort. The patients in this study were selected randomly, with the main criteria being initiation of therapy according to the RALL-2016m protocol and availability of archived bone marrow DNA at the onset of disease. Nevertheless, 15% of patients showed complete loss of genetic material in a small segment of chromosome 7 (Figure 6). Additionally, seven more patients (15%) had a monoallelic deletion in the same *TRB* locus.

The beta locus of T-cell receptor located at 7q34 includes segments V, J, D and C. During the development of T cells, the beta chain is synthesized as a result of recombination at the DNA level, connecting segment D to segment J; then segment V is attached to the DJ gene. The beta locus also includes eight trypsinogen genes, three of which encode functional proteins and five of them are pseudogenes [11]. It is likely that the result of recombination looks like a deletion when analyzing gene copy number alterations using CMA. In the majority of patients with deletions, we revealed *TRB* clonality in our study. However, we also encountered patients with deletions but no *TRB* clonality, as well as patients with clonality but without deletions. It is not clear whether the biallelic loss is caused by a rearrangement of *TRB* on one chromosome and deletion on another, or by a biallelic rearrangement leading to the production of amplicons with the same length during PCR for T-cell clonality analysis. 7q34 loss may be a favorable molecular prognostic marker associated with T-I and T-II variants of T-ALL according to our data (Appendix A). We previously conducted a similar study on a cohort of patients with B-ALL and have found a similar frequency of biallelic deletions at 9p23.1 (5 out of 36 patients, 14%). The 7q34 locus was virtually unaffected in B-ALL, with only one out of thirty-six patients (less than 3%) showing any changes [12]. We have now investigated the *TRB* clonality in this patient as well and identified a biallelic clonal rearrangement of *TRB*. We have not found any similar studies in the literature on biallelic deletions at this locus. However, the participation of the *TRB* locus in translocations in T-ALL was noted [13,14,15,16,17,18]. Raimondi notes that even early cytogenetic studies of the T-ALL genome showed non-random breakpoints in the following three clusters of T cell receptor genes *TRA*, *TRD* (14q11.2) or *TRB* (7q34). *TCR* breakpoints are present in about 30–35% of ALL [14].

During translocations, the elements of the *TRB* promoter and enhancer are combined with a relatively small number of genes encoding transcription factors, which can lead to malignancy of T cells. Chromosomal aberrations that affect *TRB* loci were among the first reported in T-ALL. Subsequently, these and other rare translocations facilitated the identification of genes that affected during T-ALL, many of which are also transcriptionally activated without any detectable chromosomal rearrangements in these loci. Several recurrent translocations like t(1;7)(p32;q34)/*TRB::TAL1*, t(7;9)(q34;q32)/*TRB::TAL2*, t(7;12)(q34;p12) *TRB::LMO3*, inversions inv(7)(p15q34) and translocations t(7;7)(p15;q34) *TRB::HOXA10*, as well as t(6;7)(q23;q34) [*TRB::MYB*] with overexpression of MYB have been reported in T-ALL [2,15]. *TRB::TLX1*, *TRB::NOTCH1*, *TRB::NKX2* are also described [16]. Molecular study of t(7;19)(q34;p13) in a pediatric patient with acute T-cell lymphoblastic leukemia led to the identification of a translocation between the *TRB* and *LYL1* loci [17]. Variants of T-ALL with inv(7) (p15q34;*TRB::TLX1*) in combination with del*CDKN2AB*/9p21 and cases of translocation of *TRB::TLX1* in combination with del*CDKN2AB*/9p21 and del *TP53*/17p13 are also described [18]. In general, the abundance of chimeric genes involving *TRB* indicates genetic instability of this locus in T-ALL. Similarly, our data support the assumption that the existence of DNA breakpoint hotspots also result in mono- and biallelic deletions covering *TRB* locus.

## 4. Materials and Methods

The study included 47 patients with Ph-negative T-ALL who received therapy at the National Medical Research Center for Hematology according to the RALL-2016m protocol from 2017 to 2023 and had available tumor DNA material at the onset of the disease. The RALL-2016m protocol is a modification of the previous RALL-2009/2016 protocols, based on the principle of low intensity and non-interruptive treatment. MRD assessment is carried out on days +70 of therapy in accordance with the protocol.

All patients included in the protocol underwent immunophenotyping, cytogenetic and molecular tests of bone marrow samples at the onset of the disease. Bone marrow cells obtained from patients during the initial examination were analyzed using G-differential chromosome staining and FISH. The FISH method was performed to excluded BCR-ABL1-positive ALL (BCR/ABL (DF) Gene Fusion Probe Detection Kit (Wuhan HealthCare Biotechnology Co., Ltd., Wuhan, China)). After a standard cytogenetic test, additional FISH tests were used to identify rearrangements of *KMT2A*/11q23 (*MLL*) (XL *MLL* plus Break Apart Probe (Metasystems, Altlussheim, Germany)), *TCRAD*/14q11 (*TCRA*/*D* Break Apart Probe (Metasystems, Altlussheim, Germany)), *TP53*/17p13 (*P53*/*CEP17* chromosome and gene anomaly detection probe Deletion Probe (Wuhan HealthCare Biotechnology Co., Ltd., Wuhan, China)), *IGH*/14q32 (IGH Break Apart Probe (Metasystems, Altlussheim, Germany)), *BCL11B*/14q32 (BCL11B Break Apart FISH Probe (Empire Genomics, New York, NY, USA)), *PDGFRA* (*PDGFRA* Gene Break Apart Probe Detection Kit, (Wuhan HealthCare Biotechnology Co., Ltd., Wuhan, China)), *PDGFRB* (*PDGFRB* Gene Break Apart Probe Detection Kit, (Wuhan HealthCare Biotechnology Co., Ltd., Wuhan, China)) genes. The karyotype and results of FISH analysis were described in accordance with the criteria of the International Cytogenomic Nomenclature ISCN, 2020 [19].

In patients who met T-ALL criteria, we performed an additional analysis to determine the immunophenotype associated with leukemia for further assessment of MRD using flow cytometry. MRD was assessed using the “different from normal” method and identification immunophenotypically immature T-cells in bone marrow [20]. Tests before 2020 were performed with BD FACSCanto II flow cytometer with 2-tube 6-color panel, which includes antibodies against CD99, CD7, CD3 (cytoplasmic and surface), CD5, CD45, CD8. Studies after 2021 were performed with BC CytoFLEX flow cytometer with single-tube 10-color panel, which includes antibodies against CD99, CD7, CD34, CD56, CD3 (cytoplasmic and surface), CD4, CD5, CD45, CD8. MRD was assessed at the end of induction (day 70) using 6- or 10-color flow cytometry of the bone marrow specimens.

CMA was carried out with Thermo Fisher Scientific (Santa Clara, CA 95151, USA) equipment using the CytoScan™ HT-CMA 96F array SNP-oligonucleotide microarray (Thermo Fisher Scientific, Waltham, MA, USA) in accordance with the manufacturer’s protocol. The analysis was performed at the “Genomed” laboratory of Molecular Pathology (Moscow, Russia). Material for analysis—DNA isolated from bone marrow cells in patients with ALL before therapy, in an amount of not less than 100 ng and not more than 200 ng with an A260/A280 ratio of not less than 1.8 and reference male DNA of a similar concentration (Thermo Fisher Scientific, Waltham, MA, USA). The scanning results were processed with the Multi Sample Viewer Software (v.1.1.0.11) and Chromosome Analysis Suite (ChAS 4.3.0.71) (Thermo Fisher Scientific, Waltham, MA, USA). Cutoff of ≥3 Mb for LOH and ≥50 kb for losses and gains was used according to Gonzales et al. [7].

PCR analysis of TRB clonality was performed according to the BIOMED-2 Concerted Action BMH4-CT98-3936 protocol [21]. Primers for multiplex PCR were synthesized by Syntol LLC (Moscow, Russia). For PCR we used the same DNA samples as for CMA.

The achievement of MRD-negative status on day 70 (end of induction) was used as surrogate endpoint to study the prognostic significance of aberrations. Categorical variables were analyzed by frequencies. We use Fisher Exact Test to test hypotheses about differences in distributions of categorical features in comparison groups. The survival analysis with Kaplan-Maier estimates and log-rank test were used for the comparison of overall survival (OS) in different groups of patients. All calculations were made using SAS 9.4.

## 5. Conclusions

According to the literature, translocations involving 7q34 are common in T-ALL. In this study, we present data on the abundance of biallelic deletions at 7q34 for the first time in T-ALL patients. The frequency of these deletions is similar to that of 9p21 deletions, which have been reported in numerous studies. We have demonstrated that the boundaries of 7q34 deletions are highly conserved and affect almost the complete beta chain of the T-cell receptor. Most of them look like the result of clonal rearrangement of TRB (6 out of 7 cases of biallelic loss in 7q34). However, it is not clear whether biallelic loss is caused by a rearrangement of the TRB on one chromosome and a deletion on another chromosome, or by a biallelic rearrangement that leads to the production of amplicons with the same length during PCR analysis for T-cell clonality. We also analyzed the association between these deletions and minimal residual disease (MRD) in T-ALL. Both biallelic deletions can be considered favorable prognostic factors, with an association with 9p21 being statistically significant and a trend for 7q34 (*p* = 0.12).

## Figures and Tables

**Figure 1 ijms-25-10482-f001:**
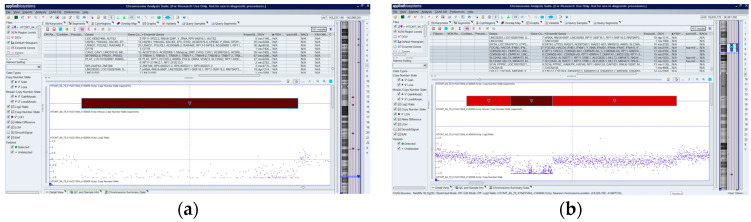
Molecular karyotype of patient # 27 with biallelic deletions of 9p21.3 (**a**) and 7q34 (**b**). Chromosomes 7 and 9 are highlighted in blue frames, and the deepred indicate the loci of the biallelic deletions. A standard cytogenetic study also revealed a translocation t(6;7)(q23;q34) in this patient, as shown in the Appendix A.

**Figure 2 ijms-25-10482-f002:**
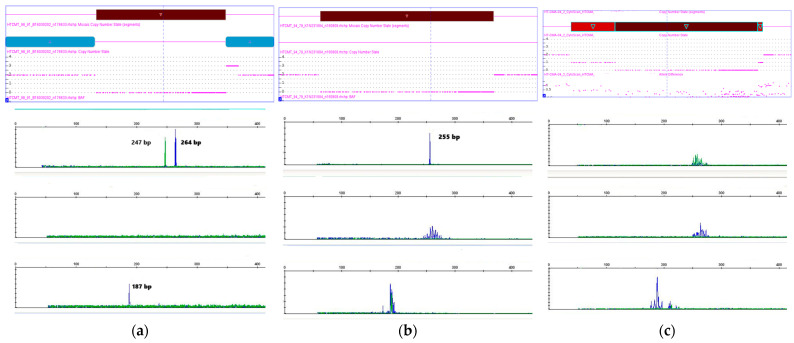
Detail views of biallelic loss 7q34 and TRB clonality assay with monoclonal peaks for patients 15 (**a**) and 27 (**b**) and polyclonal Gaussian curves for patient 44 (**c**).

**Figure 3 ijms-25-10482-f003:**
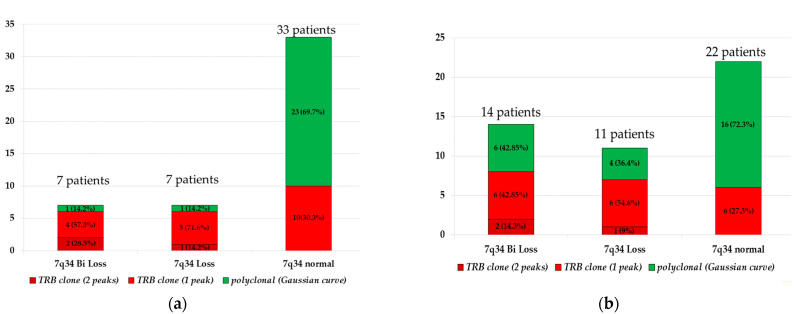
Histogram of the distribution of *TRB* clonality in groups of patients with biallelic loss of 7q34, monoallelic loss of 7q34, and intact 7q34 locus. (**a**)—CMA results with a cutoff of 50 kb for losses, (**b**)—a cutoff of 1 kb for losses. Dark red color indicates biallelic *TRB* clonality, red indicates *TRB* clonality with one monoclonal PCR peak, green indicates polyclonal samples. On the *X*-axis, the groups are arranged in the following order: with biallelic 7q34 loss, monoallelic 7q34 loss, no 7q34 lesions.

**Figure 4 ijms-25-10482-f004:**
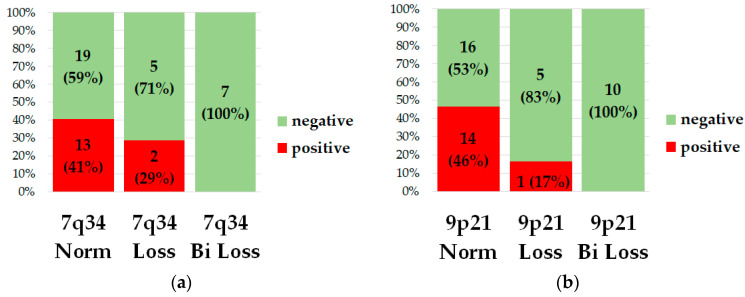
Diagrams of the distribution of the MRD status depending on the presence of loss 7q34 (**a**) or 9p23.1 (**b**). The number of patients and percentage are indicated on the bars. Green indicates MRD-negative patients, red indicates MRD-positive patients. Along the X axis the bars are arranged in the following order—without loss, with monoallelic loss, with biallelic loss.

**Figure 5 ijms-25-10482-f005:**
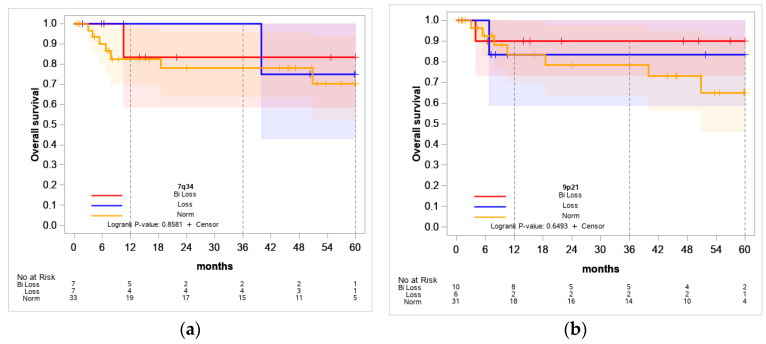
Overall survival curves of patients depending on the presence of 7q34 (**a**) and 9p21.3 (**b**) deletions. Patients with biallelic loss (red curve), monoallelic loss (blue curve) and patients without deletions (yellow curve) were analyzed. The *X*-axis indicates the time after the start of therapy (months).

**Figure 6 ijms-25-10482-f006:**
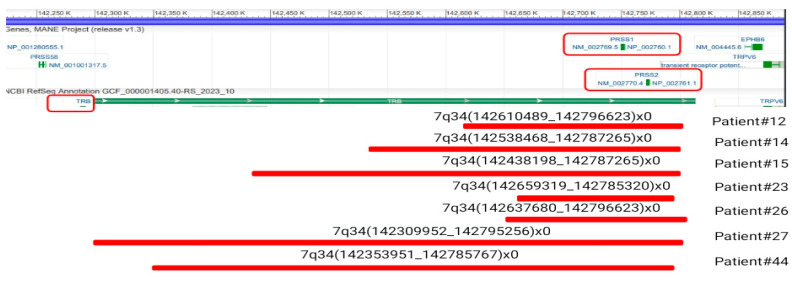
The location of biallelic deletions of 7q34 in the studied cohort of patients, relative to the *TRB* locus, is shown in red lines. This figure was adapted from https://www.ncbi.nlm.nih.gov/gene/6957 (accessed on 19 August 2024).

**Table 1 ijms-25-10482-t001:** The main characteristics of the patients.

Parameters	T-ALL (*n* = 47)
Male:Female	34:13
Age, median	33 (19–53) years
Leukocytes, 10^9^/L	44.59 (0.95–833.94)
LDH	1 207 (154–20,064)
Blast cells in peripheral blood, %	75 (0–97)
Blast cells in the bone marrow, %	86 (5.2–100)
Immunophenotype, EGIL, WHO	T-I 6 (12.7%)
	T-II 16 (34%)
	T-III 21 (44.6%)
	T-IV 1 (2.1%) MPAL(T-myelo) 3 (6.4%)
Standard cytogenetics	47
+mitosis	44 (93.6%)
−mitosis	3 (6.4%)
Karyotype	44
Normal	20 (20.0%)
Abnormal:	24 (80.0%)
CNS leukemia	10 (21.3%)
Extramedullary disease	34 (72.3%)
MRD-status (+70 day)	46
MRD+	13
MRD−	33
CR:	
After 2nd induction (+70 day)	16
Refractory disease	14
Early Death	1

T-ALL, T-cell lymphoblastic leukemia/lymphoma; LDH, lactate dehydrogenase; EGIL, European Group on Immunological Classification of Leukemia; WHO, World Health Organization; MPAL, mixed phenotype acute leukemia; CNS, central nervous system; CR, complete remission.

**Table 2 ijms-25-10482-t002:** Biallelic losses in a cohort of patients with T-ALL. The aberrations flanking the biallelic deletion are also shown. Deletions are marked in yellow, cnLOH are marked in purple. * Patient with translocation t(6;7)(q23;q34).

Patient#	Age	Gender	Immunophenotype	% of Blast Cells in Bone Marrow	9p21.3 (Start-End Position of Genomic Coordinates)	7q34 (Start-End Position of Genomic Coordinates)	12p13.2 (Start-End Position of Genomic Coordinates)	13q14.2 (Start-End Position of Genomic Coordinates)
12	47	m	T-III	80.1	9p21.3(21972814_22025494)x0	7q34(142610489_142796623)x0		
14	40	m	T-II	84.9		7q34(142538468_142787265)x0, 7q34q36.1(142792254_151451166)x1		
15	23	m	T-III	96.5		7q34(142438198_142787265)x0		
23	41	m	T-II	81.2	9p24.3p13.3(204082_33290534)x2 hmz, 9p21.3(21682143_22103814)x0	7q34(142659319_142785320)x0		
26	20	m	T-III	94.9	9p21.3(21471728_22313094)x0	7q34(142637680_142796623)x0		13q14.2(48411504_48496038)x0
27	40	m	T-III	87.5	9p22.1p21.3(19703574_20908514)x1, 9p21.3(20914896_22056500)x0, 9p21.3(22061616_25458801)x1	7q34(142309952_142795256)x0 *		
28	42	f	T-III	61.5	9p21.3(21052665_21865843)x1, 9p21.3(21866502_21996864)x0, 9p21.3(21998037_25552602)x1			
30	54	f	T-II	86			12p13.2(11592806_11983676)x0	
36	40	m	T-II	38	9p24.3p21.3(204083_21635941)x1 [0.76], 9p21.3(20272224_22647015)x0 [0.5], 9p21.3p13.1(22681778_38780194)x1 [0.76]			
38	21	f	T-III	59.3	9p24.3p21.3(204082_21905380)x1, 9p21.3(21909980_22233512)x0, 9p21.3p11.2(22238596_40880243)x1			
43	33	f	T-III	89.3	9p24.3p21.3(203862_20710799)x1, 9p21.3(20719868_22088261)x0, 9p21.3p11.2(22283154_41008724)x1			
44	30	f	T-III	71		7q34(142353951_142785767)x0		
45	29	m	T-II	94.8	9p22.3p21.3(14432343_22130515)x1, 9p21.3p21.2(22137086_25938431)x0, 9p21.2q13(25944009_67986968)x1			
46	23	m	T-III	66	9p21.3(21720109_22502487)x0			

## Data Availability

Data are contained within the article and Appendix A.

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
