# Peer review of "Biallelic Loss of 7q34 (TRB) and 9p21.3 (CDKN2A/2B) in Adult Ph-Negative Acute T-Lymphoblastic Leukemia"

_ijms, 2024, doi:10.3390/ijms251910482_

Round 1

Reviewer 1 Report

Comments and Suggestions for Authors

In this work conducted on 47 Ph-negative T-ALL adult patients the authors observed 14 cases of biallelic deletion of 9p21.3 or 7q34, alone or combined, all MRD-negative at day 70 of RALL 2016 clinical protocol. This suggests a favorable prognosis in the presence of such biallelic deletions. The work is interesting, and the results are discussed in detail.

Ten patients had a biallelic deletion of 9p21.3, involving the tumor suppressor CDKN2A gene. This apparently paradoxical result has been discussed by the authors citing the work of Van Vlierberghe et al, who observed that the homozygous deletion of CDKN2A/CDKN2B in cortical and mature T-ALLs is associated with a favorable outcome. In this regard, the authors should indicate the immunophenotype of the observed cases with biallelic deletion of 9p21.3.

Minor points:

In figure 3 the curve of patients with monoallelic loss is in blue, while in the legend it is indicated to be in yellow.

In the text, genes should always be indicated in italics.

Comments on the Quality of English Language

A minor editing of the English form is required.

Author Response

Dear Reviewer, Thank you for your positive and constructive comments on our manuscript. We appreciate your feedback and will take it into consideration in the revision process.

Comments 1: [In figure 3 the curve of patients with monoallelic loss is in blue, while in the legend it is indicated to be in yellow.]

Response 1: [Fixed]

Comments 2: [In the text, genes should always be indicated in italics.]

Response 2: [

We have carefully checked all the instances of human gene naming in the manuscript and changed them to italics.

We also included some speculations about TRB clonality and its association with 7q34 loss in the "Results" section.

(Results 2.2. Lines 135-165)]

Reviewer 2 Report

Comments and Suggestions for Authors

The manuscript presents an interesting observation of biallelic deletions at the 7q34 locus in T-ALL patients, an event not previously reported. However, there are several aspects of the study that warrant further scrutiny and caution in the interpretation of the results.

- While the authors utilized chromosomal microarray analysis (CMA) to detect biallelic deletions, it is widely acknowledged that CMA, despite its utility, can sometimes yield false positives or artifacts, especially in complex genomic regions like the TRB locus. Thenovelty of the reported biallelic 7q34 deletion, therefore, requires confirmation through alternative methodologies such as fluorescence in situ hybridization (FISH), targeted PCR,or DNA sequencing to validate the findings and rule out technical artifacts.

- The Authors suggest a potential association between biallelic deletions of 7q34 and 9p21.3. While this is an intriguing hypothesis, the small sample size (only four patients with both deletions) significantly limits the statistical power of this observation. Furthermore, without detailed information regarding the treatment regimens of these patients, it is difficult to assess the true clinical relevance of this association. The therapeutic context plays a crucial role in the prognostic evaluation, and its omission represents a critical gap in the study.

- The Authors propose that the presence of these biallelic deletions may be associated with a favorable prognosis, as suggested by the MRD-negative status at day 70. However, given the limited number of cases and the absence of comprehensive treatment details, these prognostic claims should be approached with caution. Prognosis in T-ALL is multifactorial, and while genetic aberrations are important, they must be considered alongside treatment variables that have not been fully accounted for in this study.

Author Response

Dear reviewer, Thank you very much for your detailed review and thorough analysis of our manuscript. We appreciate your feedback and are confident that the changes we have made based on your suggestions should move our work closer to publication.

Comments 1: [While the authors utilized chromosomal microarray analysis (CMA) to detect biallelic deletions, it is widely acknowledged that CMA, despite its utility, can sometimes yield false positives or artifacts, especially in complex genomic regions like the TRB locus. Thenovelty of the reported biallelic 7q34 deletion, therefore, requires confirmation through alternative methodologies such as fluorescence in situ hybridization (FISH), targeted PCR,or DNA sequencing to validate the findings and rule out technical artifacts.]

Response 1: [We performed TRB clonality PCR analysis on all 47 patients, and the results are included in the "Results" section. 2.2. Association between 7q34 loss and clonal TRB rearrangement (Lines 135-165).]

Comments 2: [The Authors suggest a potential association between biallelic deletions of 7q34 and 9p21.3. While this is an intriguing hypothesis, the small sample size (only four patients with both deletions) significantly limits the statistical power of this observation. Furthermore, without detailed information regarding the treatment regimens of these patients, it is difficult to assess the true clinical relevance of this association. The therapeutic context plays a crucial role in the prognostic evaluation, and its omission represents a critical gap in the study.]

Response 2: [We believe that a possible connection between two simultaneous events is simply the mathematical likelihood of the occurrence of the two most common independent events at the same time. However, due to the small sample size, we were unable to conduct a statistical analysis. Therefore, we simply reported the frequency of occurrences of two events. We have also included a diagram of the RALL 2016 clinical protocol (Supplementary Figure S1) and detailed information on the individual therapy regimens in Supplementary Table S1.]

Comments 3: [The Authors propose that the presence of these biallelic deletions may be associated with a favorable prognosis, as suggested by the MRD-negative status at day 70. However, given the limited number of cases and the absence of comprehensive treatment details, these prognostic claims should be approached with caution. Prognosis in T-ALL is multifactorial, and while genetic aberrations are important, they must be considered alongside treatment variables that have not been fully accounted for in this study.]

Response 3: [We have justified the choice of the surrogate endpoint - the MOB status at day 70 - in the introduction. (Lines 60-70).
We have also added the following paragraph «Therefore, it is difficult to estimate the overall survival rate (Figure 5) accurately. We did not find any significant differences in patients' outcomes depending on the presence of deletions. This may be due to the fact that a number of patients received targeted therapy or allogeneic HSCT in the case of MRD positivity» to the "Results"  Section 2.3 (Lines 190-195). We understand that a more in-depth analysis of a larger sample of patients is required to assess the association between the described lesions and the prognosis of the disease. This analysis should take into account different therapy regimens as well as other molecular factors. We were able to conduct only a preliminary and very tentative analysis of the relationship between the most common T-ALL molecular abnormalities and clinical outcomes.]

Round 2

Reviewer 2 Report

Comments and Suggestions for Authors

The authors improved the research as requested by inserting more figures (fig. 2 and 4) and making more clear that one already present (former fig. 2, now 3).

They increased even the supplementary files